# Systematized Event-Aware Learning for Multi-Object Tracking

**Hyemin Lee**[1]

**Daijin Kim**[1]

[1]Department of Computer Science and Engineering, Pohang University of Science and Technology, Pohang, Korea

## Abstract

We propose an end-to-end online multi-object tracking (MOT) framework with a systematized event-aware loss, which is designed to control possible occurrences in an online MOT situation and compel the tracker to take appropriate actions when such events occur. Training samples from real candidates using a simulation tracker are generated, and a systematized event-aware association matrix is constructed for every frame to enable the tracker to learn the ideal action in a running environment. Several experiments, including ablation studies on various public MOT benchmark datasets, are conducted. The experimental results verify that each event affecting the tracking measure can be controlled, and the proposed method presents optimal results compared with recent state-of-the-art MOT methods.

## 1 INTRODUCTION

Multi-object tracking (MOT) is a fundamental computer vision task that has been applied to video surveillance, human–computer interaction, advanced driver assistance systems, and autonomous driving. Deep convolutional neural networks (CNNs) have been successfully applied to MOT methods, particularly in the use of deep features, for accurate candidate associations by learning similarity measures between the features. Notwithstanding the benefits of CNNs, a decline in MOT performance occurs owing to certain events, such as target missing, target disappearance, false positive detection, and new target appearance.

In several methods, the deep neural network is trained to prevent these abnormal events; however, existing methods only train the subnetworks of the MOT framework, and the training samples and loss cannot directly reduce the occurrence of such abnormal events. Even if each part of an MOT

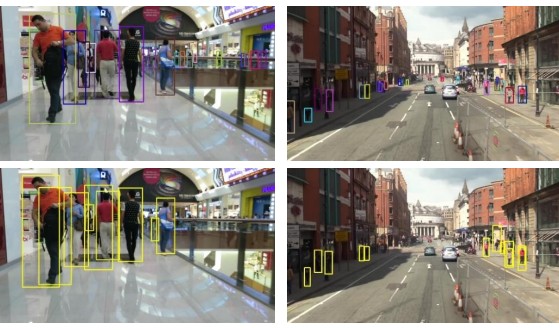

Figure 1: Discrepancy between GT bounding box (top) and the public detections provided by MOT benchmark datasets (bottom): the GT contains boxes for an occluded target, and the detection box is missing or includes false positives in the on provided detections.

network is trained effectively, the trained network is not guaranteed to be optimized for a real running environment. This is because most MOT methods train their subnetworks without using a unified network architecture or unified multi-task loss, which enables end-to-end learning. Another reason is that MOT networks are typically trained using frame-by-frame ground-truth (GT) information, without considering the temporal information and potential diversity that can occur during real MOT running process. We found the discrepancy between public detections provided by the MOT benchmark and the GT information, as depicted in Figure 1. The real environment contains incomplete detections that may cause occlusion and false alarms; inappropriate target initialization and tracking termination are other potential problems.

To address these problems, we propose an end-to-end online MOT framework with a systematized event-aware loss. The systematized event-aware loss is designed to control every possible abnormal event that can occur in an online MOT situation and to make the tracker take appropriate actions when such an event occurs. We systematically analyze

*Accepted for the 38th Conference on Uncertainty in Artificial Intelligence* (UAI 2022).

the entire case of assignment between the target and candidate, and we categorize the case into meaningful events. We first categorize the targets and detection candidates into five event cases: tracking success, target missing, target disappearance, false positive, and new target appearance. Each target and candidate is assigned an event-aware loss that can measure how well the tracker takes the ability of the tracker to take an appropriate action for each target. The loss can be back-propagated from the association layer to the feature extraction layer, which includes the entire MOT process in a unified framework. Training samples from real actual tracking candidates using a simulation tracker are generated and a systematized event-aware association matrix is constructed for every frame to optimize the tracker in a real environment.

To demonstrate that our proposed method can effectively improve tracking performance, we compare it with state-of-the-art trackers and perform an MOT benchmark evaluation using the MOT2015, MOT2016, and MOT2017 datasets. For all of the datasets, the tracking performance is improved by the proposed method, demonstrating its efficacy. We also conduct ablation studies to identify how and where the proposed network improves tracking performance, including various loss settings and affinity definition. The main contributions of this study are as follows.

- We propose a systematized event-aware loss that can successfully train the entire MOT network to deal with abnormal events.

- We train the MOT networks in an end-to-end manner by using proposed event-aware loss and reduce the potential error caused by FNs, FPs, and tracking termination.

- We perform extensive experiments to demonstrate and verify how the proposed method improves the tracking performance.

## 2 RELATED WORK

In this section, we briefly introduce the MOT problem and related studies. MOT algorithms can be categorized as performing offline or online tracking and as using private or public detection. Our methods focus on online trackers that use public detection.

**Offline trackers.** Offline trackers utilize the entire frame and detection bounding boxes in a batch to predict a trajectory Kim and Kim [2016]. Therefore, they focus on global optimization methods, such as network flow Dehghan et al. [2015] and multiple hypotheses Chu et al. [2016]. Offline trackers are capable of using future and past information simultaneously; however, one limitation is that they cannot be used in real-time applications, such as real-time surveillance or automatic warning systems.

**Online trackers.** Online trackers can only access current and past frames, and not future frames. Therefore, they make a decision at every frame, and the trajectory reported in the current frame cannot be corrected after being reported. The Hungarian method is frequently used to associate targets because the linear assignment problem must be solved for every frame Chen et al. [2017], Fagot-Bouquet et al. [2015]. Deep CNN features have been successfully utilized in online MOT methods. For example, Chu et al. [2017], Zhu et al. [2018] used deep CNN features to associate candidates and long short-term memory (LSTM) to enhance discrimination features by utilizing temporal information. Recently, the MOT using graph convolutional neural network is proposed Papakis et al. [2020].

**MOT with a single-object tracker (SOT).** The limitation of association-based tracking-by-detection is that a missing target cannot be associated with a candidate when there are no suitable candidates. To overcome this limitation, some MOT methods, such as Chu et al. [2019], Zhu et al. [2018], Yin et al. [2020], adopt a SOT to complement missing detection by using the SOT prediction as a new candidate. The methods in Chu et al. [2017] initialized all detections as SOT targets and unified an SOT module into a framework. Some MOT methods, such as Yin et al. [2020], Lee et al. [2020], integrate SOT and affinity calculations in a unified network.

**MOT with an object detector.** Some trackers adopt private detector to solve MOT problem. Typically, these methods are referred to as one-shot methods because the tracker contains its own detection network inside the MOT network. Zhang et al. [2020], Zhou et al. [2020]. For fair comparison, in the MOT benchmark, an MOT using public detections cannot use its own detector, which can considerably affect the performance of the MOT. This means that the internal detector cannot generate extra target candidates, except for those provided by the public detections. Recently, Tracktor Bergmann et al. [2019] used a regression and classification layer of detection networks without a region proposal network to solve the MOT problem. This method enables the MOT to take advantage of object detectors, which classify objects as background or target. Regression is performed from the previous position to the current frame to determine the location of the target.

**Optimization using matrix-wise loss.** The authors of Sun et al. [2019] proposed a deep affinity network to perform optimization using matrix loss. This method first proposes matrix-wise loss to train the network in single frames, but it does not include the motion model or SOT so that it has weakness when detection is missing. In DeepMOT Xu et al. [2020], a deep Hungarian network (DHN) comprised of a bidirectional RNN that performs the Hungarian algorithm was proposed. The DHN functions as a bridge between the affinity calculation and association, thus enabling direct loss backpropagation after association.

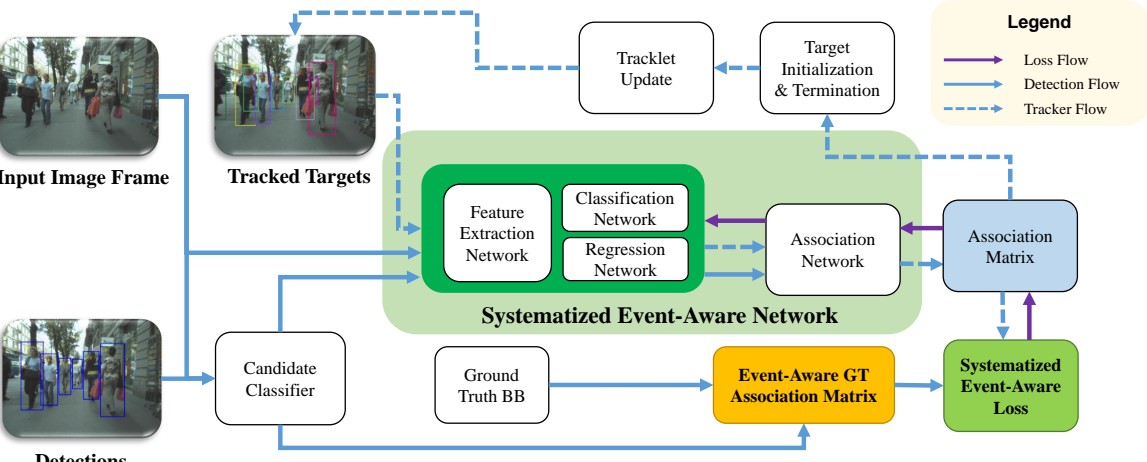

Figure 2: Overview of tracking method trained using systematized event-aware loss. The tracker can be trained in end-to-end manner to reduce the occurrence of abnormal event and perform appropriate action if such an event occurs

## 3 PROPOSED METHOD

In the inference phase, the proposed MOT method considers an image frame, detection boxes, and targets from the previous frame as inputs. First, the provided detection boxes are filtered using classifiers to filter the FP. The remaining detection boxes are used as target candidates to be initialized as new targets. Second, each tracking target independently tracks the target, and the affinity between the tracked target and candidate is calculated. Using the calculated affinity matrix, we inference the association matrix through DHN. Based on this association matrix, the tracker finds the new target to be initialized and determines whether to maintain the track or terminate the tracking. The entire association process is performed in an association network, and tracking management is performed using the output of the association network. In the training phase, a ground-truth event-aware association matrix is constructed by observing two consecutive frames, and each target and candidate is categorized based on the matching cases. Using the event-aware association matrix, we obtain a systematized event-aware loss, and the entire network is trained to prevent abnormal events and take appropriate action when certain events occur. Figure 2 shows the overall process of the proposed method.

### 3.1 CANDIDATE CLASSIFIER

Several MOT methods adopt candidate classifiers to filter detection bounding boxes because the provided detection bounding boxes contain several FP boxes. FP candidates are a significant problem in MOT because they generate continuous false trajectories when initialized as new targets, making association confusing. These FP candidates can be filtered by applying an additional classifier, as discussed in Long et al. [2018], Bergmann et al. [2019]. The authors of

Bergmann et al. [2019] adapted the classification network of a faster R-CNN detector Ren et al. [2015] with ResNet-101 He et al. [2016] as a backbone network. The pooled features corresponding to each detection candidate were classified into the background and object classes. The difference between candidate classifier and object detection is that the tracker cannot generate additional candidates that are not included in the given detection candidate. This means that the tracker can only classify the target from the given detection, not from the anchor boxes.

In contrast to existing candidate classifiers, which are pre-processing networks trained independently using 2-class samples, in our network, the candidate classifier is integrated into a unified network architecture. Therefore, we train the network in an end-to-end manner using real candidate boxes obtained from the simulation tracker, which can be optimized using the proposed event-aware loss.

In our proposed method, the set of filtered candidates in the $t$-th frame is denoted as $D_t = \{d_t\}$, where each detection box is denoted as $d_t = \{d_t^x, d_t^y, d_t^w, d_t^h\}$. The features of an input image frame $I_t$ are extracted using the backbone network, and the classification scores corresponding to each candidate are calculated by applying a classification network for each candidate. Then, candidates with classification scores higher than the threshold are used in the next tracking step.

### 3.2 EVENT CATEGORIZATION USING TARGET STATE TABLE

Before defining the event-aware loss, all event cases are categorized based on the target state table obtained from consecutive frames. First, we find target matching between ground truth boxes and detection boxes on frames $t-1$ and

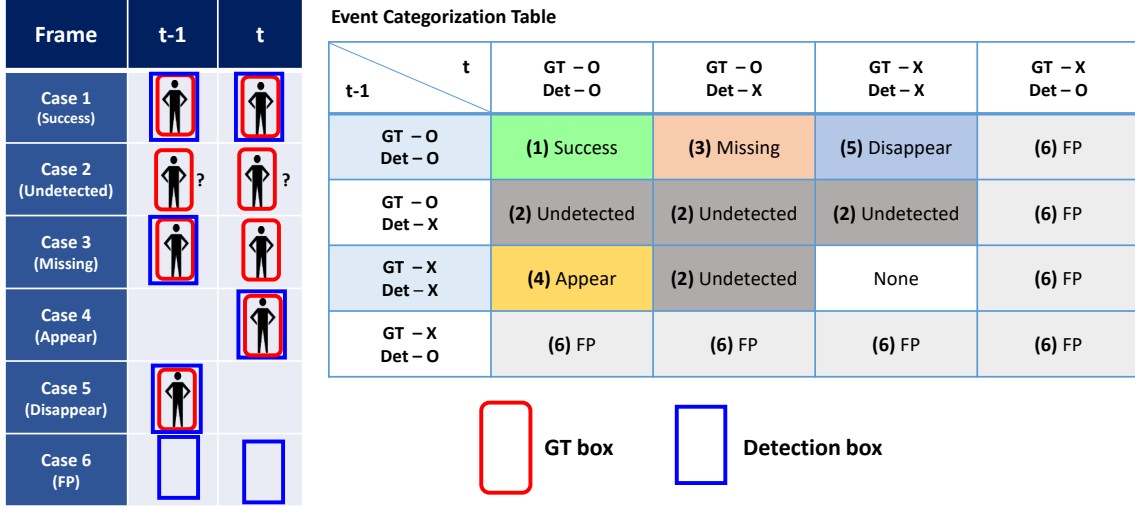

Figure 3: Process of assigning event for each detection box by associating it with ground-truth boxes. If the ground-truth box is undetected in the previous frame, we do not consider such a box. If the box is detected in the previous frame and undetected in the current frame, it is considered a missing target. If the target appears in the current frame, it is considered a new target appearance. The disappeared target in the ground-truth at the current frame is considered a disappeared target. The remaining detections that are not associated whether in the previous and current frame are treated as false positives (FPs).

$t$, respectively. The set of detections at $t-1$, and $D_{t-1}$ and set of ground-truth boxes $G_{t-1}$ are associated using the IoU measure, and the ID from the ground truth is assigned to the matched detection boxes. The matched target is assigned as the detected target, and the unmatched target is assigned as an undetected target. To distinguish them from undetected boxes, we denote the GT boxes whose IDs do not appear in the current frame as the absent state. Next, we perform the same process for the detection and GT at frame $t$ with $D_t$ and $G_t$ and determine whether the targets are detected or undetected. Subsequently, we obtain the target state table of the $t-1$ frame and $t$ frame for all targets.

We precisely analyze the entire case of states and categorize all possible states into event groups.

**Case 1. Success**    If the box is detected on both $t-1$ and $t$, we assign the same ID to each detected box, and categorize them into a successful case.

**Case 2. Undetected**    If the box is undetected in the previous frame, we do not consider such a box because it denotes that the ground-truth box cannot be detected by the detector, and this target cannot be initialized.

**Case 3. Target missing**    If the box is detected in the previous frame and undetected in the current frame, it is assigned "target missing." In this case, we add the GT bounding box corresponding to the missing target into the association candidate at $t$. This is because the tracker should be trained to

find the proper location when the detection loss (FN) occurs.

**Case 4. New target appearance**    If the target appears and is detected only in the current frame, it is assigned to a new target appearance.

**Case 5. Disappeared target**    If the target ID exists in $t-1$ and not in $t$, we assign that target as the disappeared target.

**Case 6. False positives**    The remaining detection candidates that are not matched on both $t-1$ and $t$ are treated as false positives.

To train the tracker regression network more precisely, we replace the detection boxes at $t$ with the corresponding GT boxes. Consequently, even if the target is initialized using noisy detection box, it could refine the bounding box during tracking. Figure 3 shows the whole possible case of target state and corresponding event categorization.

### 3.3 GROUND-TRUTH EVENT-AWARE ASSOCIATION MATRIX CONSTRUCTION

In this section, we describe the construction of a ground-truth event-aware association matrix using a simulation tracker and event-assigned bounding boxes. In the simulation tracker environment, we assume that $t-1$ is the initial frame, and initialize the tracking targets $X_{t-1}$ with $D_{t-1}$. Each initialized tracking target follows the assigned

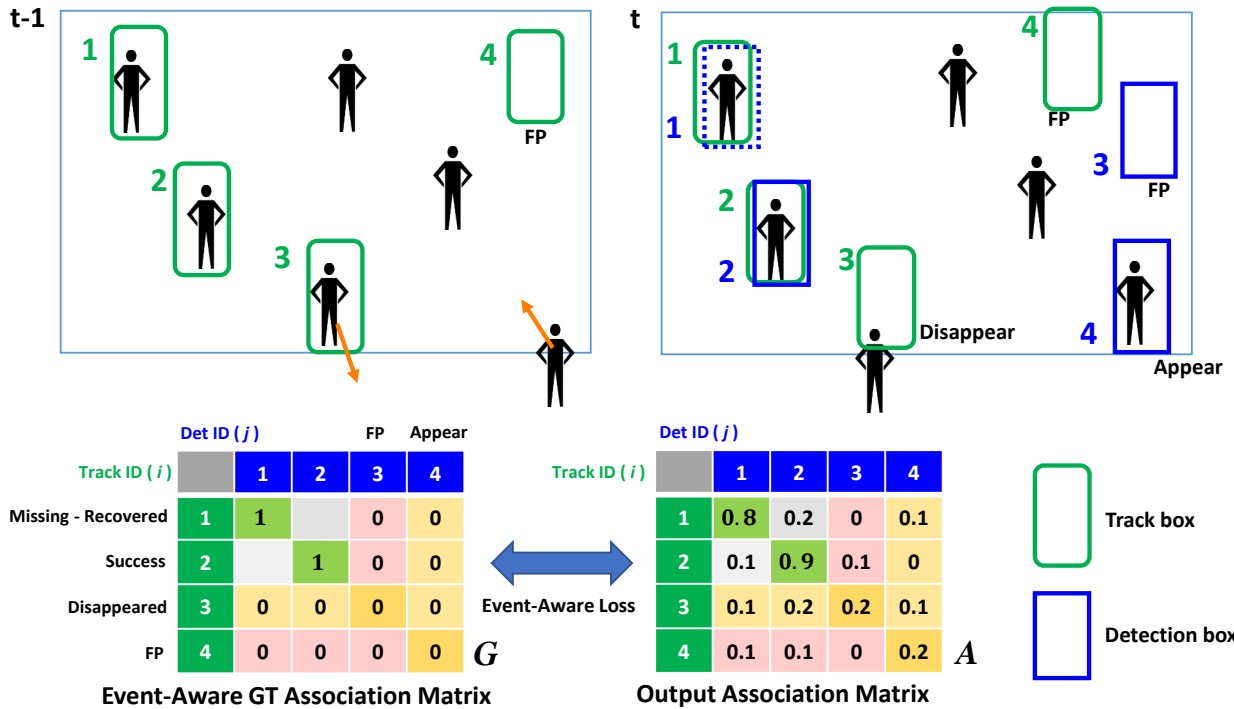

Figure 4: Process of constructing an event-aware ground-truth association matrix from categorized ground-truth bounding box and given categorized detection bounding boxes.

label of the previous detection box $D_{t-1}$. Subsequently, we perform motion tracking for all targets using a regression network and obtain new track bounding boxes $X_t$. Using the obtained new track positions, we construct an affinity matrix $F_t$ between $D_t$ and $X_t$ using the following affinity function:

$$f(d_t, x_t) = \frac{GIoU(d_t, x_t) + (\psi(d_t) \star \psi(x_t))}{2} \quad (1)$$

where the affinity feature encoding layer is denoted as $\psi(\cdot)$ and the cosine similarity is denoted as $\star$.

In contrast to Bergmann et al. [2019], a combination of appearance affinity using the Siamese network and position affinity using generalized intersection over union (GIoU) Rezatofighi et al. [2019] is used.

The affinity matrix $F_t$ is fed into the DHN to perform the association task. Consequently, we obtain the soft association matrix $A_t$. Each column and row of $A_t$ represents the tracked targets $x_t$ and detections $d_t$. Because we have pre-categorized all targets and detections, we can assign the ground-truth value on each element of matrix.

We propose a principle that is effective for tracking scenarios. If the target and detection are matched successfully, the association value should be higher than that of any other target. Therefore, we assign one to both the ground-truth table for matched target and detection and for the missed target. This is because we the ground truth boxes for the missed target have been added in the previous step to make the tracker regress into an appropriate box in frame $t$. If the detection is categorized into FP detection or newly appeared target, the box should not be associated with any targets. Subsequently, we assign the values of the entire column corresponding to these detections to zero. Similarly, from the target perspective, if the target is initialized with FP or the disappeared target, the target should not be associated with any detections. We assign the values of the entire row corresponding to these target IDs to zero. Figure 4 illustrates the process of constructing a ground-truth event-aware association matrix. In the next section, we describe the processes involved in training the MOT network using this constructed ground-truth association matrix.

### 3.4 END-TO-END TRAINING USING SYSTEMATIZED EVENT-AWARE LOSS

We define the systematized event-aware loss by comparing the ground-truth association matrix $G$ and the obtained association matrix $A$. First, the matching loss $L_m$ is defined as the difference between the GT value and the obtained association value for matched target. We denote the set of matched targets as $M$, and the matching loss is defined as

$$L_m = \sum_{d \in M, x \in M} \frac{|G_{ij} - A_{ij}|}{|M|} \quad (2)$$

where $i$ denotes the column index of $d$ and $j$ denotes the row index of $x$. Next, the FP loss is defined as the normalized distance between false positive targets $X_{fp}$ and detections $D_{fp}$, disappeared target $E$, and newly appeared target $N$.

The FP loss is defined as

$$L_{fp} = \sum_{d \in (D_{fp} \cup N)} \frac{||G_i - A_i||_2}{|D_{fp}| + |N|} + \sum_{x \in (X_{fp} \cup E)} \frac{||G_j - A_j||_2}{|X_{fp}| + |E|}. \tag{3}$$

The FN loss is defined as the difference of the GT value on lost target $S$ as

$$L_{fn} = \sum_{d \in S, x \in S} \frac{|G_{ij} - A_{ij}|}{|S|}. \tag{4}$$

Finally, the overall loss, which is the sum of these losses is defined as :

$$L = L_m + \alpha L_{fp} + \beta L_{fn} \tag{5}$$

where $\alpha$ and $\beta$ are the weighting factor.

In summary, we categorize all detections and targets into event groups using a target state table. Based on the categorization, all detection boxes should be included in one of the four following sets: the matched detection and targets pairs $M$, FP detections $D_{fp}$, newly appeared targets $N$, or lost targets and recovered detection pairs $S$. In additions, all targets should be included in one of the four following sets: the matched targets and detection pairs $M$, disappeared targets $E$, false positive tracking targets $X_{fp}$, or lost targets $S$. $d$ denotes an element in the detection set, and $x$ denotes an element in the target set. $G$ is the GT association matrix, and $A$ is the predicted association matrix. The size of matrix $G$ and $X$ is $|D|x|X|$ where $|D|$ is the number of whole detections and $|X|$ is the number of whole targets. The subscript $i$ denotes the column index term of $d$ in $G$ and $A$. The subscript $j$ denotes the row index term of $x$ in $G$ and $A$. The goal of our training is to make $A$ exactly same with $G$. Because the number of targets and detections are different for each frame, the size of the matrix is also different. We want to assign different weight to each element in the matrix to categorize each element into event groups, separate the detection and targets.

With the help of the deep hungarian network proposed in Xu et al. [2020], the proposed systematized event-aware loss can be back-propagated through the affinity network, classification and regression network, and feature extraction network, which contains the entire MOT network. This is to ensure that the tracker can be optimized to reflect the usefulness of the end-to-end training.

## 3.5   TRAINING DATA GENERATION

We randomly cropped the training sequences into a single sample sequences with overlap. Each sample contains the

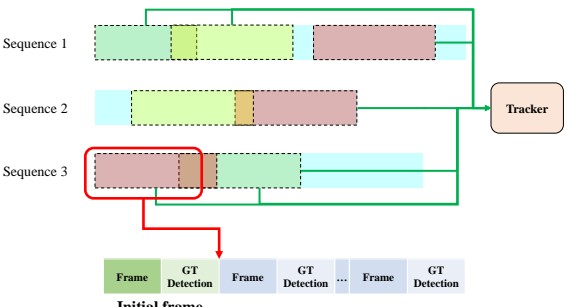

Figure 5: Illustration of the generation of the simulation data: the training sequences are composed of the frame image, corresponding GT bounding boxes, and provided detection boxes. We randomly crop the sequences to produce training data, which is used to training the tracker.

consecutive image frames, ground truth bounding boxes, and detection bounding boxes. The training sequences includes MOT2015 and MOT2017 training sets. For each sample, we initialize the simulation tracker using given initial frame information of each sample, and run the simulation tracker until end of the sample sequence. The frame length of training data is set to random values between 10 to 20 frames. It is important to note that all tracker are initialized with the given detections of first frame in one sample as same with inference phase. The generation of simulation data is illustrated in Figure 5.

## 3.6   NETWORK ARCHITECTURE

We used an R-FCN architecture with SqueezeNet as the backbone network for the MOTDT baseline, and used the Faster R-CNN detector with ResNet-101 and feature pyramid networks (FPNs) Lin et al. [2017] as the backbone network for the Tracktor baseline. The regression network performs bounding box regression based on the location of previous frames to the location of subsequent frames. This network comprises a 1x1 convolutional network and 2 fully connected layers (FCs) and generates the relevant offset of each bounding box: $dx, dy, dw$, and $dh$. The association network for MOTDT uses GoogLeNet for the association features, and the association network for the Tracktor baseline was implemented based on the Siamese CNN architecture trained on TriNet Hermans et al. [2017] using ResNet-50. We followed the same tracking management strategy baseline tracker excluding the association and training steps. The detailed network architectures are illustrated on Figure6.

## 4   EXPERIMENTS

We conducted extensive experiments to determine the effectiveness of the proposed network on three MOT benchmark

Table 1: Tracking Performance on the MOT2015 benchmark test set. Best in bold.

| Method | MOTA↑ | IDF1↑ | MT↑ | ML↓ | FP↓ | FN↓ | IDS↓ |
|---|---|---|---|---|---|---|---|
| SCEA (Hong Yoon et al. [2016]) | 29.1 | 37.2 | 8.9 | 47.3 | 6060 | 36912 | 604 |
| MDP (Xiang et al. [2015]) | 30.3 | 44.7 | 13.0 | 38.4 | 9717 | 32422 | 680 |
| AP (Chen et al. [2017]) | 38.5 | 47.1 | 8.7 | 37.4 | **4006** | 33203 | 586 |
| KCF (Chu et al. [2019]) | 38.9 | 44.5 | 16.6 | 31.6 | 7321 | 29501 | 720 |
| DeepMOT (Xu et al. [2020]) | 44.1 | 46.0 | 17.2 | 26.6 | 6085 | 26917 | 1347 |
| GNNMatch (Papakis et al. [2020]) | 46.7 | 43.2 | 21.8 | 28.2 | 6643 | 25311 | 820 |
| MOTDT (Long et al. [2018]) | 33.1 | 44.3 | 9.1 | 46.2 | 6806 | 36226 | 616 |
| SEAT (MOTDT) | 35.3 | 45.8 | 12.9 | 45.9 | 8217 | 29209 | **472** |
| Tracktor++ (Bergmann et al. [2019]) | 44.1 | 46.7 | 18.0 | **26.2** | 6477 | 26577 | 1318 |
| SEAT (Tracktor++) | 45.2 | 47.4 | 20.8 | 27.0 | 6943 | 25373 | 1339 |
| Tracktor++v2 (Bergmann et al. [2019]) | 46.6 | 48.3 | 18.2 | 27.9 | 4624 | 26896 | 1290 |
| SEAT (Tracktor++v2) | **47.0** | **48.4** | **22.2** | 26.5 | 6654 | **24632** | 1275 |

Table 2: Tracking Performance on the MOT2016 benchmark test set. Best in bold.

| Method | MOTA↑ | IDF1↑ | MT↑ | ML↓ | FP↓ | FN↓ | IDS↓ |
|---|---|---|---|---|---|---|---|
| STAM (Chu et al. [2017]) | 46.0 | 50.0 | 14.6 | 43.6 | 6895 | 91117 | **473** |
| DMAN (Zhu et al. [2018]) | 46.1 | 54.8 | 17.4 | 42.6 | 7909 | 89874 | 532 |
| AMIR (Sadeghian et al. [2017]) | 47.2 | 46.3 | 14.0 | 41.6 | 2681 | 92856 | 774 |
| KCF (Chu et al. [2019]) | 48.8 | 47.2 | 15.8 | 38.1 | 5875 | 86567 | 906 |
| UMA (Yin et al. [2020]) | 50.5 | 52.8 | 17.8 | **33.7** | 7587 | 81924 | 685 |
| DeepMOT (Xu et al. [2020]) | 54.8 | 53.4 | 19.1 | 37.0 | 2955 | 78765 | 645 |
| GSM (Liu et al. [2020]) | 57.0 | **58.2** | 22.0 | 34.5 | 4332 | 73573 | 475 |
| GNNMatch (Papakis et al. [2020]) | 57.2 | 55.0 | 22.9 | 34.0 | 3905 | 73493 | 559 |
| MOTDT (Long et al. [2018]) | 47.6 | 50.9 | 15.2 | 38.3 | 9253 | 85431 | 792 |
| SEAT (MOTDT) | 48.2 | 50.7 | 14.1 | 37.1 | 8869 | 84784 | 838 |
| Tracktor++ (Bergmann et al. [2019]) | 54.4 | 52.5 | 19.0 | 36.9 | 3280 | 79149 | 682 |
| SEAT (Tracktor++) | 54.9 | 54.9 | 21.3 | 36.1 | 4683 | 76953 | 590 |
| Tracktor++v2 (Bergmann et al. [2019]) | 56.2 | 54.9 | 20.7 | 35.8 | **2394** | 76844 | 617 |
| SEAT (Tracktor++v2) | **57.2** | 57.6 | **24.5** | 33.9 | 4208 | **73215** | 574 |

Table 3: Tracking Performance on the MOT2017 benchmark test set. Best in bold.

| Method | MOTA↑ | IDF1↑ | MT↑ | ML↓ | FP↓ | FN↓ | IDS↓ |
|---|---|---|---|---|---|---|---|
| PHD_GSDL (Fu et al. [2017]) | 48.0 | 49.6 | 17.1 | 35.6 | 23199 | 265954 | 3988 |
| AM_ADM (Yoon et al. [2018]) | 48.1 | 52.1 | 13.4 | 39.7 | 25061 | 265495 | 2214 |
| DMAN (Zhu et al. [2018]) | 48.2 | 55.7 | 19.3 | 38.3 | 26218 | 263608 | 2194 |
| HAM_SADF (Yoon et al. [2018]) | 48.3 | 51.1 | 17.1 | 41.7 | 20967 | 269038 | 1871 |
| FAMNet (Chu and Ling [2019]) | 52.0 | 48.7 | 19.1 | 33.4 | 14138 | 253616 | 5318 |
| UMA (Yin et al. [2020]) | 53.1 | 54.4 | 21.5 | **31.8** | 22893 | 239534 | 2251 |
| DeepMOT (Xu et al. [2020]) | 53.7 | 53.8 | 19.4 | 36.6 | 11731 | 247447 | 1947 |
| GSM (Liu et al. [2020]) | 56.4 | 57.8 | 22.2 | 34.5 | 14379 | 230174 | **1485** |
| GNNMatch (Papakis et al. [2020]) | 57.0 | 56.1 | 23.3 | 34.6 | 12283 | 228242 | 1957 |
| MOTDT (Long et al. [2018]) | 50.9 | 52.7 | 17.5 | 35.7 | 24069 | 250768 | 2474 |
| SEAT (MOTDT) | 51.7 | 51.8 | 19.7 | 33.2 | 30755 | 238813 | 3223 |
| Tracktor++ (Bergmann et al. [2019]) | 53.5 | 52.3 | 19.5 | 36.6 | 12201 | 248047 | 2072 |
| SEAT (Tracktor++) | 54.5 | 55.0 | 23.4 | 34.3 | 19453 | 235500 | 1975 |
| Tracktor++v2 (Bergmann et al. [2019]) | 56.3 | 55.1 | 21.1 | 35.3 | **8866** | 235449 | 1987 |
| SEAT (Tracktor++v2) | **57.3** | **58.2** | **26.2** | 33.4 | 15832 | **223051** | 1873 |

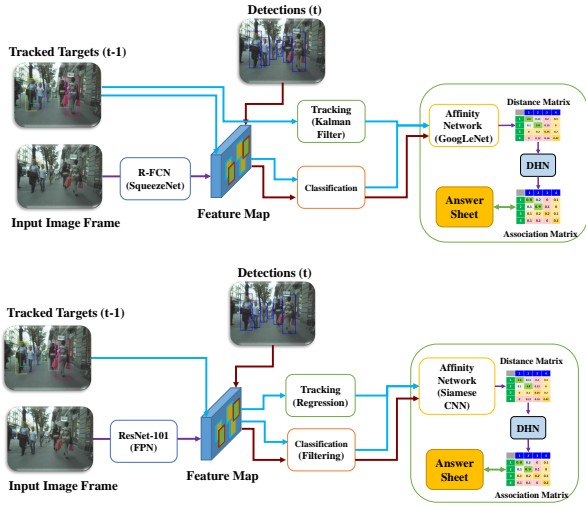

Figure 6: Network architectures of proposed methods.

datasets: MOT2015, MOT2016, and MOT2017 Milan et al. [2016]. The results of the other trackers and the proposed method were evaluated using the official MOT challenge benchmark score board[1].

## 4.1 IMPLEMENTATION DETAILS

The proposed method was implemented using PyTorch and tested on a six-core Intel i7@3.60 GHz CPU and NVIDIA Titan Xp GPU environment. We used the Faster R-CNN detector with ResNet-101 as the backbone network for the baseline tracker. The minimum threshold value for filtering candidates was set to 0.4. The network was trained using stochastic gradient descent over 30 epochs with learning rates ranging from $10^{-3}$ to $10^{-5}$. We generated training samples from the 2D MOT2015 and MOT2017 training sets and split them seven-fold to train the network. The classification threshold for target initialization was set to 0.3 and the maximum lost time for termination was set to 40 frames.

The training requires 4.6 GB GPU memory storage and approximately 22-hours for 30 training epochs using a single Titan XP GPU. The inference requires a maximum of 900 MB memory storage for the backbone and target appearance model. More implementation details are provided in the supplementary material.

## 4.2 EVALUATION ON MOT BENCHMARKS

The proposed method was evaluated on MOT2015, MOT2016, and MOT2017 test datasets from an official

[1]https://motchallenge.net

website. We adopted the CLEAR MOT metrics Bernardin and Stiefelhagen [2008] to evaluate the performance of the tracker on the MOT datasets and compared it with other state-of-the-art trackers. The representative metric was the multiple object tracking accuracy (MOTA), which reflects the false negatives (FN), false positives (FP), and identity switches (IDS). Other metrics have also been reported, including identity F1 scores (IDF1), percentage of mostly tracked targets (MT), and mostly lost targets (ML). We denoted our proposed tracker as SEAT(Online Systematized Event-Aware Tracker).

The 2D MOT2015 test dataset consists of 11 video sequences obtained from various scenes with ACF detection results. The tracking performance evaluated on MOT2015 test dataset is listed in Table 1. The MOT2016 test dataset contains seven videos that were entirely disjointed with the training set with DPM detection results. The tracking performance evaluated on MOT2016 test dataset is listed in Table 2. The MOT2017 test dataset contains the same video sequences as the MOT2016 dataset; however, different detections were provided. This dataset focused on evaluating trackers based on various detection results. Three types of detectors were used in this dataset: DPM, SDP, and Faster-RCNN. The results of the MOT2017 test dataset are listed in Table 3. We evaluated the proposed tracker using the same network model and hyperparameters throughout the testing process.

The proposed method also achieves excellent results in terms of MOTA, ML, and FN compared to existing state-of-the-art MOT methods, including offline methods that can utilize global optimization. In particular, our method significantly reduced FNs. The experimental results demonstrate the remarkable performance of the proposed SEAT.

Note that we did not report on the trackers using an external detector and trained on supplemental training datasets in addition to MOT datasets. This is because the tracking performance can be dominated by the performance for the detector in such cases.

## 4.3 DISCUSSION OF TIME CONSUMPTION

The strength of our proposed network is that we can share all subnetworks for all targets and detections. The only additional burdens of additional targets is cosine similarity calculation and the Hungarian algorithm, whose time complexity is $O(n^3)$. By the speed measure, the MOT17 contains a frame that has 8 to 70 people, but the inference speed of our method is only from 7.2 Hz to 9.5 Hz, which is scalable. Note that the baseline ran on 8.6 Hz on average, and the proposed method ran on 8.1 Hz on average, demonstrating a minor additional time burden.

Table 4: Ablation study based on various tracker versions and loss definitions.

| Baseline | Training loss | MOTA↑ | IDF1↑ | MT↑ | ML↓ | FP↓ | FN↓ | IDS↓ |
|---|---|---|---|---|---|---|---|---|
| *Tracktor++* | Base | 67.4 | 66.7 | 39.9 | 17.2 | 848 | 35412 | 332 |
| | $L_m$ | 67.9 | 67.7 | 41.5 | 17.3 | 909 | 34715 | 390 |
| | $L_m + L_{fp}$ | 67.1 | 67.5 | 38.4 | 17.2 | **741** | 35711 | 464 |
| | $L_m + L_{fn}$ | 68.4 | 69.1 | **44.1** | **16.1** | 2259 | **32883** | 348 |
| | $L_m + L_{fn} + L_{fp}$ | **68.8** | **69.2** | 44.0 | 17.0 | 1511 | 33223 | **318** |
| *Tracktor++v2* | Base | 68.2 | 68.5 | 42.5 | 17.3 | 1036 | 34304 | 361 |
| | $L_m$ | 67.8 | 66.0 | 43.7 | **16.1** | 2771 | 32955 | 425 |
| | $L_m + L_{fp}$ | 68.1 | 68.1 | 42.3 | 17.4 | **1007** | 34432 | 371 |
| | $L_m + L_{fn}$ | 69.0 | 70.1 | 45.0 | 16.7 | 1904 | 32522 | 305 |
| | $L_m + L_{fn} + L_{fp}$ | **69.3** | **71.8** | **45.6** | 16.5 | 2108 | **32078** | **297** |

Table 5: Ablation study based on various affinity function.

| Function | MOTA↑ | IDF1↑ | MT↑ | ML↓ | FP↓ | FN↓ | IDS↓ |
|---|---|---|---|---|---|---|---|
| Appearance | 66.5 | 66.3 | 37.9 | 17.2 | **1168** | 36004 | 884 |
| IoU | 68.9 | 70.3 | 43.9 | 16.3 | 1737 | 32832 | 336 |
| GIoU | 69.1 | 71.3 | **46.1** | **16.3** | 2381 | **31970** | 300 |
| IoU + Appearance | 69.2 | 71.0 | 45.4 | 16.5 | 1995 | 32272 | 303 |
| GIoU + Appearance | **69.3** | **71.8** | 45.6 | 16.5 | 2108 | 32078 | **297** |

## 4.4 ABLATION STUDY

We performed additional experiments for ablation studies by using various versions of the proposed tracker to determine the effect of loss on tracking performance and verify the effectiveness of the proposed approach. The experiments for the ablation studies were performed on a subset of the MOT2017 training dataset that was not used in the training phase. This is because the corresponding testing dataset did not provide ground truth labels for validation. We evaluated the SDP sequences from the MOT2017 dataset.

**Ablation Study of Various Loss Definition.** We trained the two versions of the trackers using the baseline association network and the versatile affinity network. The ablation study showed the effect of loss on each term of the evaluation results. When we used only the matching loss $L_m$, the performance was poor and further degraded at the baseline. The addition of FP loss $L_{fp}$ can reduce the number of false positives; however, the entire performance is not optimized. In contrast, the addition of FN loss $L_{fn}$ can reduce the number of false negatives and simultaneously improve the entire performance. This is because the influence of FN is relatively high. Finally, using all the systematized event-aware loss, we could obtain optimal results compared with the baseline tracker. The results are shown in Table 4. Additional experimental results using various weighting factors for each loss term are reported in the supplementary material.

**Ablation Study of Various Affinity Functions.** We also tested the proposed tracker using various affinity functions to compute affinity matrix. Note that we fixed the trained weight of the affinity network and performed tracking evaluation by applying a different affinity function during test time. The result showed that appearance features alone are not sufficient to calculate the affinity. The results of affinity using GIoU were relatively good compared to the one using IoU. The results are shown in Table 5.

## 5 CONCLUSIONS

We developed a novel end-to-end online MOT framework with a systematized event-aware loss that is trained to prevent possible abnormal events and compel the tracker to take appropriate actions when such events occur. The proposed loss reflects a systematized tracking event, including missing targets, target disappearance, false alarms, and new target appearances. The experimental results showed that the proposed method outperformed state-of-the-art online MOT methods, and systematized event-aware loss can successfully train the entire MOT network to deal with the occurrence of abnormal events.

## ACKNOWLEGEMENT

This work was supported by Institute of Information & communications Technology Planning & Evaluation(IITP) grant funded by the Korea government(MSIT) (No.B0101-15-0266, Development of High Performance Visual BigData Discovery Platform for Large-Scale Realtime Data Analysis), (No.2017-0-00897, Development of Object Detection and Recognition for Intelligent Vehicles) and (No.2018-0-01290, Development of an Open Dataset and Cognitive Processing Technology for the Recognition of Features Derived From Unstructured Human Motions Used in Self-driving Cars)

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
