# OpenReview forum: "Systematized Event-Aware Learning for Multi-Object Tracking"
_auai.org/UAI/2022/Conference — UAI 2022 Poster_

### Official Review · Reviewer_JrQG · 2022-04-11

**Q2(1) Originality/Novelty:** 3
**Q2(2) Significance/Impact:** 3
**Q2(3) Correctness/Technical Quality:** 2
**Q2(6) Clarity Of Writing:** 3
**Q6 Overall Score:** 6
**Q8 Confidence In Your Score:** 4

**Q1 Summary And Contributions:**

In this paper, a systematized event-aware loss is designed to deal with these problems in different situations of multi-target tracking in practical scenarios, and it is proved that this end-to-end training method can effectively improve tracking performance in multiple data sets.

**Q2 Assessment Of The Paper:**

More detailed information regarding each of these aspects is given below:

**Q2(4) Quality Of Experiments (Optional):**

2: Fair: The experimental evaluation is weak: important baselines are missing, or the results do not adequately support the main claims.

**Q2(5) Reproducibility:**

3: Good: Key resources (e.g., proofs, code, data) are available and key details (e.g., proofs, experimental setup) are sufficiently well-described for competent researchers to confidently reproduce the main results.

**Q3 Main Strengths:**

This paper analyzes the problems in object tracking in detail, divides them into several cases that can be detected, and then uses different loss training tracking models. This method provides some novel solutions for multi-object tracking.

**Q4 Main Weakness:**

The end-to-end training method proposed in this paper takes into account different tracing cases, but it seems to add more additional time consumption. This paper needs more explanation and analysis on these problems.

**Q5 Detailed Comments To The Authors:**

(1)Use punctuation marks after formulas, then use a capital letter to start a sentence in English.
(2)References for different methods should be added to the table presented with experimental results.
(3)Time complexity should be included in the experimental discussion section. Please describe in detail the impact of the proposed method on model training and test phases.

**Q7 Justification For Your Score:**

In this paper, an end-to-end tracking model training method is proposed by analyzing the specific problems of multi-object tracking. The tracking performance is improved obviously by designing corresponding loss functions for different cases.
At the same time, it is worth note that the impact of this training method on the time complexity.

**Q9 Complying With Reviewing Instructions:**

1: Yes.

---

### Official Review · Reviewer_va6u · 2022-04-12

**Q2(1) Originality/Novelty:** 3
**Q2(2) Significance/Impact:** 2
**Q2(3) Correctness/Technical Quality:** 3
**Q2(6) Clarity Of Writing:** 3
**Q6 Overall Score:** 6
**Q8 Confidence In Your Score:** 3

**Q1 Summary And Contributions:**

The authors present a new on-line multiple object tracking methodology. The key idea of the paper is the identify particularly difficult events that cause significant problems to the performance of traditional MoT methods, such as new targets or disappearance of established targets. The authors suggest a composite total loss that combines several different losses, in order to reflect these difficult situations. An assignment problem (hungarian network) is applied to drive the loss calculation.

**Q2 Assessment Of The Paper:**

More detailed information regarding each of these aspects is given below:

**Q2(4) Quality Of Experiments (Optional):**

3: Good: The experimental evaluation is adequate, and the results convincingly support the main claims.

**Q2(5) Reproducibility:**

1: Poor: Key details (e.g., proof sketches, experimental setup) are incomplete/unclear, or key resources (e.g., proofs, code, data) are unavailable.

**Q3 Main Strengths:**

- The approach appears novel. I particularly liked the approach to systematise categorically the types of (sub)-loses based on actual problems in real applications. The authors try to generalise to a reasonable number of such rare event categories, however, it is not clear if these offer complete coverage of the possible situations.
- The assignment problem of events to categories is relaxed by a Hungarian network which offers efficiency. Finally, the authors show that their complete apparatus is learnable and thus can be easily back-propagated.
- The results are convincing.
- I find the paper easy to read and the discussion clear.

**Q4 Main Weakness:**

Although the discussion is indeed clear, I find that a large number of details are missing, which would be crucial for others to follow, verify and extend this work. I would have appreciated an appendix (online one would suffice).

**Q5 Detailed Comments To The Authors:**

1) The paper is well written. I spotted a typo: last paragraph, right column, first page.

2) How did the authors decide to chose the parameters (Implementation details) and how do these affect the overall performance? What is the estimated effort to calibrate these for other data sets? Are these sensitive?

3) How about training (and inference) costs? Could the authors offer any details? How is the method scaling with the number of targets?

**Q7 Justification For Your Score:**

Please see above.

**Q9 Complying With Reviewing Instructions:**

1: Yes.

---

### Official Review · Reviewer_cHMa · 2022-04-15

**Q2(1) Originality/Novelty:** 2
**Q2(2) Significance/Impact:** 2
**Q2(3) Correctness/Technical Quality:** 2
**Q2(6) Clarity Of Writing:** 1
**Q6 Overall Score:** 3
**Q8 Confidence In Your Score:** 2

**Q1 Summary And Contributions:**

The paper proposes a multi-object tracking framework by generating associative matrices which make a finer classification of events during tracking. The proposed approach achieve SOTA performance on MOT benchmarks.

**Q2 Assessment Of The Paper:**

More detailed information regarding each of these aspects is given below:

**Q2(4) Quality Of Experiments (Optional):**

2: Fair: The experimental evaluation is weak: important baselines are missing, or the results do not adequately support the main claims.

**Q2(5) Reproducibility:**

2: Fair: Key resources (e.g., proofs, code, data) are unavailable but key details (e.g., proof sketches, experimental setup) are sufficiently well-described for an expert to confidently reproduce the main results.

**Q3 Main Strengths:**

Strength:

- the event-based tracking idea is good.
- the method achieved SOTA on the standard benchmarks.

**Q4 Main Weakness:**

The presentation is not clear. I can hardly understand the formulas. See comments for detail.

**Q5 Detailed Comments To The Authors:**

The paper has flaws in the the presentation, making me feel difficult to understand its idea. For example, I do not find the place to define the regression network. Besides, the description of formulas in 3.4 seems very confusing, thus I can not understand what those formulas mean. - - For example, what does $x$ and $d$ mean in equation (2)? They are never defined before.
- The meaning of $M$ is also obscure. Is it a set of pairs or a set of bboxes or a matrix?
- In the next paragraph, the distance is defined between “four terms”, which does not make sense to me.
- The subscript $i, j$ in equation (3) is not defined.
- Then,the FN loss is defined as the difference of the GT value. But how can we compute the difference of a single value?

**Q7 Justification For Your Score:**

The paper's writing is tough to follow, and it is hard to make a fair evaluation based on its current form. I suggest rejecting the submission.

**Q9 Complying With Reviewing Instructions:**

1: Yes.

---

### Decision · Program_Chairs · 2022-05-15

**Decision:**

Accept (Poster)

**Comment:**

Meta Review: This paper introduces an end-to-end online multi-object tracking methodology.  It receives three reviews; with two weak accepts and one reject.  The reviewers believe the proposed approach is novel, with convincing experimental results.  The main concerns include unclear presentation, missing details, and, failure to discuss computational complexity on the proposed method.  The authors adequately address some of concerns during rebuttal.